# How transferable are features in deep neural networks?

**Jason Yosinski,**[1]   **Jeff Clune,**[2]   **Yoshua Bengio,**[3]   **and  Hod Lipson**[4]
[1] Dept. Computer Science,  Cornell University
[2] Dept. Computer Science,  University of Wyoming
[3] Dept. Computer Science & Operations Research,  University of Montreal
[4] Dept. Mechanical & Aerospace Engineering,  Cornell University

## Abstract

Many deep neural networks trained on natural images exhibit a curious phenomenon in common: on the first layer they learn features similar to Gabor filters and color blobs. Such first-layer features appear not to be *specific* to a particular dataset or task, but *general* in that they are applicable to many datasets and tasks. Features must eventually transition from general to specific by the last layer of the network, but this transition has not been studied extensively. In this paper we experimentally quantify the generality versus specificity of neurons in each layer of a deep convolutional neural network and report a few surprising results. Transferability is negatively affected by two distinct issues: (1) the specialization of higher layer neurons to their original task at the expense of performance on the target task, which was expected, and (2) optimization difficulties related to splitting networks between co-adapted neurons, which was not expected. In an example network trained on ImageNet, we demonstrate that either of these two issues may dominate, depending on whether features are transferred from the bottom, middle, or top of the network. We also document that the transferability of features decreases as the distance between the base task and target task increases, but that transferring features even from distant tasks can be better than using random features. A final surprising result is that initializing a network with transferred features from almost any number of layers can produce a boost to generalization that lingers even after fine-tuning to the target dataset.

## 1   Introduction

Modern deep neural networks exhibit a curious phenomenon: when trained on images, they all tend to learn first-layer features that resemble either Gabor filters or color blobs. The appearance of these filters is so common that obtaining anything else on a natural image dataset causes suspicion of poorly chosen hyperparameters or a software bug. This phenomenon occurs not only for different datasets, but even with very different training objectives, including supervised image classification (Krizhevsky *et al.*, 2012), unsupervised density learning (Lee *et al.*, 2009), and unsupervised learning of sparse representations (Le *et al.*, 2011).

Because finding these standard features on the first layer seems to occur regardless of the exact cost function and natural image dataset, we call these first-layer features *general*. On the other hand, we know that the features computed by the last layer of a trained network must depend greatly on the chosen dataset and task. For example, in a network with an N-dimensional softmax output layer that has been successfully trained toward a supervised classification objective, each output unit will be specific to a particular class. We thus call the last-layer features *specific*. These are intuitive notions of *general* and *specific* for which we will provide more rigorous definitions below. If first-layer

features are general and last-layer features are specific, then there must be a transition from general to specific somewhere in the network. This observation raises a few questions:

- Can we quantify the degree to which a particular layer is general or specific?
- Does the transition occur suddenly at a single layer, or is it spread out over several layers?
- Where does this transition take place: near the first, middle, or last layer of the network?

We are interested in the answers to these questions because, to the extent that features within a network are general, we will be able to use them for *transfer learning* (Caruana, 1995; Bengio *et al.*, 2011; Bengio, 2011). In transfer learning, we first train a *base* network on a base dataset and task, and then we repurpose the learned features, or *transfer* them, to a second *target* network to be trained on a target dataset and task. This process will tend to work if the features are general, meaning suitable to both base and target tasks, instead of specific to the base task.

When the target dataset is significantly smaller than the base dataset, transfer learning can be a powerful tool to enable training a large target network without overfitting; Recent studies have taken advantage of this fact to obtain state-of-the-art results when transferring from higher layers (Donahue *et al.*, 2013a; Zeiler and Fergus, 2013; Sermanet *et al.*, 2014), collectively suggesting that these layers of neural networks do indeed compute features that are fairly general. These results further emphasize the importance of studying the exact nature and extent of this generality.

The usual transfer learning approach is to train a base network and then copy its first $n$ layers to the first $n$ layers of a target network. The remaining layers of the target network are then randomly initialized and trained toward the target task. One can choose to backpropagate the errors from the new task into the base (copied) features to *fine-tune* them to the new task, or the transferred feature layers can be left *frozen*, meaning that they do not change during training on the new task. The choice of whether or not to fine-tune the first $n$ layers of the target network depends on the size of the target dataset and the number of parameters in the first $n$ layers. If the target dataset is small and the number of parameters is large, fine-tuning may result in overfitting, so the features are often left frozen. On the other hand, if the target dataset is large or the number of parameters is small, so that overfitting is not a problem, then the base features can be fine-tuned to the new task to improve performance. Of course, if the target dataset is very large, there would be little need to transfer because the lower level filters could just be learned from scratch on the target dataset. We compare results from each of these two techniques — fine-tuned features or frozen features — in the following sections.

In this paper we make several contributions:

1. We define a way to quantify the degree to which a particular layer is general or specific, namely, how well features at that layer transfer from one task to another (Section 2). We then train pairs of convolutional neural networks on the ImageNet dataset and characterize the layer-by-layer transition from general to specific (Section 4), which yields the following four results.

2. We experimentally show two separate issues that cause performance degradation when using transferred features without fine-tuning: (i) the specificity of the features themselves, and (ii) optimization difficulties due to splitting the base network between co-adapted neurons on neighboring layers. We show how each of these two effects can dominate at different layers of the network. (Section 4.1)

3. We quantify how the performance benefits of transferring features decreases the more dissimilar the base task and target task are. (Section 4.2)

4. On the relatively large ImageNet dataset, we find lower performance than has been previously reported for smaller datasets (Jarrett *et al.*, 2009) when using features computed from random lower-layer weights vs. trained weights. We compare random weights to transferred weights—both frozen and fine-tuned—and find the transferred weights perform better. (Section 4.3)

5. Finally, we find that initializing a network with transferred features from almost any number of layers can produce a boost to generalization performance after fine-tuning to a new dataset. This is particularly surprising because the effect of having seen the first dataset persists even after extensive fine-tuning. (Section 4.1)

## 2   Generality vs. Specificity Measured as Transfer Performance

We have noted the curious tendency of Gabor filters and color blobs to show up in the first layer of neural networks trained on natural images. In this study, we define the degree of generality of a set of features learned on task A as the extent to which the features can be used for another task B. It is important to note that this definition depends on the similarity between A and B. We create pairs of classification tasks A and B by constructing pairs of non-overlapping subsets of the ImageNet dataset.[1] These subsets can be chosen to be similar to or different from each other.

To create tasks A and B, we randomly split the 1000 ImageNet classes into two groups each containing 500 classes and approximately half of the data, or about 645,000 examples each. We train one eight-layer convolutional network on A and another on B. These networks, which we call baseA and baseB, are shown in the top two rows of Figure 1. We then choose a layer $n$ from $\{1, 2, \ldots, 7\}$ and train several new networks. In the following explanation and in Figure 1, we use layer $n = 3$ as the example layer chosen. First, we define and train the following two networks:

- A *selffer* network B3B: the first 3 layers are copied from baseB and frozen. The five higher layers (4–8) are initialized randomly and trained on dataset B. This network is a control for the next transfer network. (Figure 1, row 3)

- A *transfer* network A3B: the first 3 layers are copied from baseA and frozen. The five higher layers (4–8) are initialized randomly and trained toward dataset B. Intuitively, here we copy the first 3 layers from a network trained on dataset A and then learn higher layer features on top of them to classify a new target dataset B. If A3B performs as well as baseB, there is evidence that the third-layer features are general, at least with respect to B. If performance suffers, there is evidence that the third-layer features are specific to A. (Figure 1, row 4)

We repeated this process for all $n$ in $\{1, 2, \ldots, 7\}$[2] and in both directions (i.e. AnB and BnA). In the above two networks, the transferred layers are *frozen*. We also create versions of the above two networks where the transferred layers are *fine-tuned*:

- A *selffer* network B3B$^+$: just like B3B, but where all layers learn.

- A *transfer* network A3B$^+$: just like A3B, but where all layers learn.

To create base and target datasets that are similar to each other, we randomly assign half of the 1000 ImageNet classes to A and half to B. ImageNet contains clusters of similar classes, particularly dogs and cats, like these 13 classes from the biological family *Felidae*: {*tabby cat, tiger cat, Persian cat, Siamese cat, Egyptian cat, mountain lion, lynx, leopard, snow leopard, jaguar, lion, tiger, cheetah*}. On average, A and B will each contain approximately 6 or 7 of these felid classes, meaning that base networks trained on each dataset will have features at all levels that help classify some types of felids. When generalizing to the other dataset, we would expect that the new high-level felid detectors trained on top of old low-level felid detectors would work well. Thus A and B are similar when created by randomly assigning classes to each, and we expect that transferred features will perform better than when A and B are less similar.

Fortunately, in ImageNet we are also provided with a hierarchy of parent classes. This information allowed us to create a special split of the dataset into two halves that are as semantically different from each other as possible: with dataset A containing only *man-made* entities and B containing *natural* entities. The split is not quite even, with 551 classes in the man-made group and 449 in the natural group. Further details of this split and the classes in each half are given in the supplementary material. In Section 4.2 we will show that features transfer more poorly (i.e. they are more specific) when the datasets are less similar.

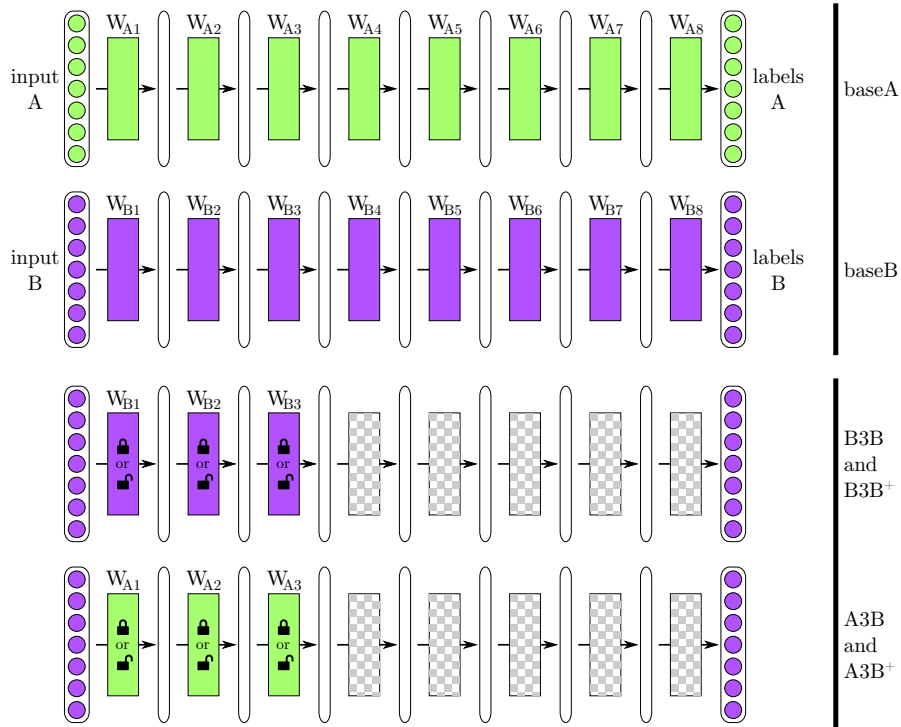

Figure 1: Overview of the experimental treatments and controls. *Top two rows:* The base networks are trained using standard supervised backprop on only half of the ImageNet dataset (first row: A half, second row: B half). The labeled rectangles (e.g. $W_{A1}$) represent the weight vector learned for that layer, with the color indicating which dataset the layer was originally trained on. The vertical, ellipsoidal bars between weight vectors represent the activations of the network at each layer. *Third row:* In the *selffer* network control, the first $n$ weight layers of the network (in this example, $n = 3$) are copied from a base network (e.g. one trained on dataset B), the upper $8 - n$ layers are randomly initialized, and then the entire network is trained on that same dataset (in this example, dataset B). The first $n$ layers are either locked during training ("frozen" selffer treatment B3B) or allowed to learn ("fine-tuned" selffer treatment B3B$^+$). This treatment reveals the occurrence of *fragile co-adaptation*, when neurons on neighboring layers co-adapt during training in such a way that cannot be rediscovered when one layer is frozen. *Fourth row:* The *transfer* network experimental treatment is the same as the selffer treatment, except that the first $n$ layers are copied from a network trained on one dataset (e.g. A) and then the entire network is trained on the *other* dataset (e.g. B). This treatment tests the extent to which the features on layer $n$ are general or specific.

## 3 Experimental Setup

Since Krizhevsky *et al.* (2012) won the ImageNet 2012 competition, there has been much interest and work toward tweaking hyperparameters of large convolutional models. However, in this study we aim not to maximize absolute performance, but rather to study transfer results on a well-known architecture. We use the reference implementation provided by Caffe (Jia *et al.*, 2014) so that our results will be comparable, extensible, and useful to a large number of researchers. Further details of the training setup (learning rates, etc.) are given in the supplementary material, and code and parameter files to reproduce these experiments are available at http://yosinski.com/transfer.

## 4 Results and Discussion

We performed three sets of experiments. The main experiment has random A/B splits and is discussed in Section 4.1. Section 4.2 presents an experiment with the man-made/natural split. Section 4.3 describes an experiment with random weights.

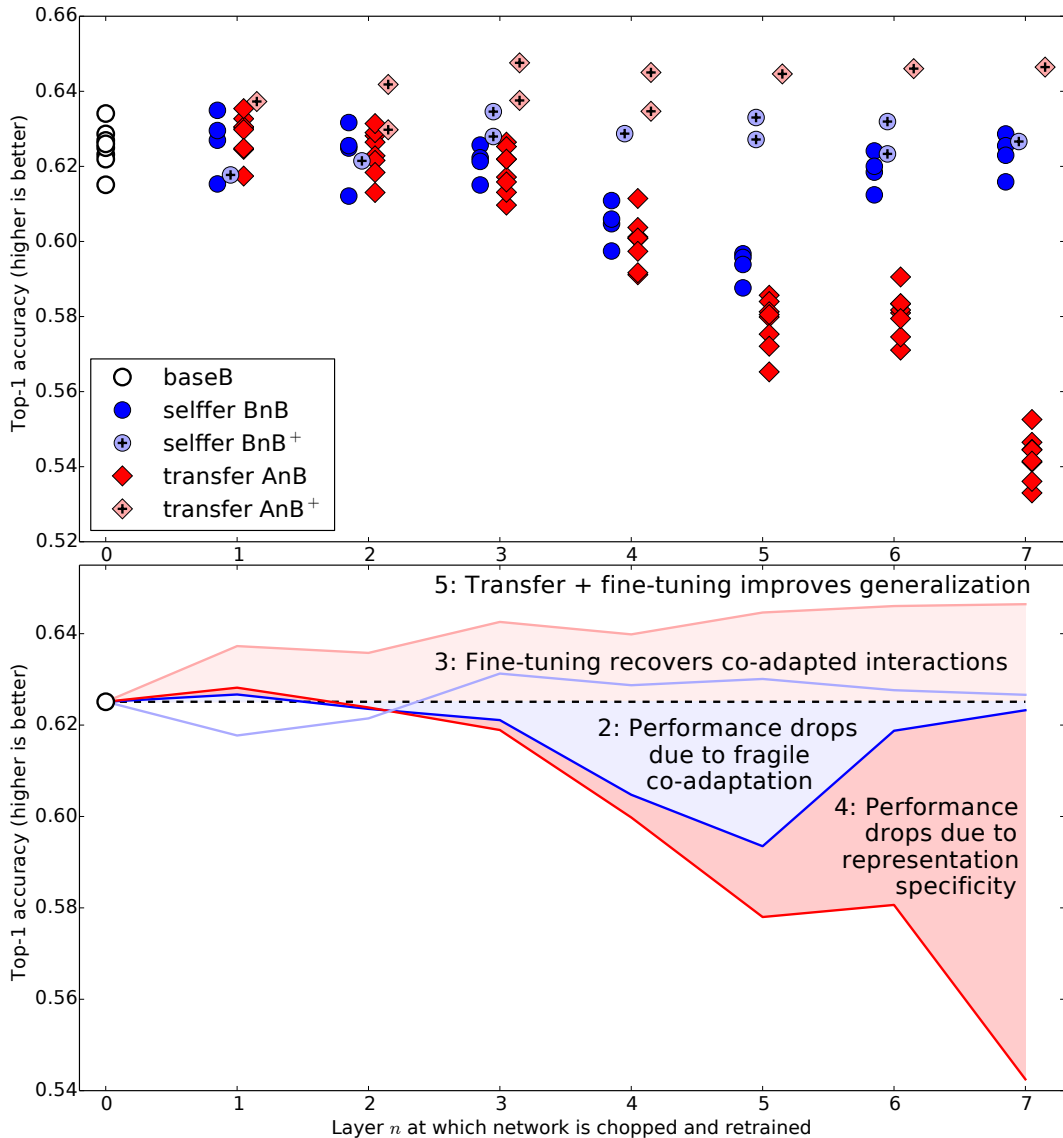

Figure 2: The results from this paper's main experiment. *Top*: Each marker in the figure represents the average accuracy over the validation set for a trained network. The white circles above $n = 0$ represent the accuracy of baseB. There are eight points, because we tested on four separate random A/B splits. Each dark blue dot represents a BnB network. Light blue points represent $BnB^+$ networks, or fine-tuned versions of BnB. Dark red diamonds are AnB networks, and light red diamonds are the fine-tuned $AnB^+$ versions. Points are shifted slightly left or right for visual clarity. *Bottom*: Lines connecting the means of each treatment. Numbered descriptions above each line refer to which interpretation from Section 4.1 applies.

## 4.1  Similar Datasets: Random A/B splits

The results of all A/B transfer learning experiments on randomly split (i.e. similar) datasets are shown[3] in Figure 2. The results yield many different conclusions. In each of the following interpretations, we compare the performance to the base case (white circles and dotted line in Figure 2).

1. The white baseB circles show that a network trained to classify a random subset of 500 classes attains a top-1 accuracy of 0.625, or 37.5% error. This error is lower than the 42.5% top-1 error attained on the 1000-class network. While error might have been higher because the network is trained on only half of the data, which could lead to more overfitting, the net result is that error is lower because there are only 500 classes, so there are only half as many ways to make mistakes.

2. The dark blue BnB points show a curious behavior. As expected, performance at layer one is the same as the baseB points. That is, if we learn eight layers of features, save the first layer of learned Gabor features and color blobs, reinitialize the whole network, and retrain it toward the same task, it does just as well. This result also holds true for layer 2. However, layers 3, 4, 5, and 6, particularly 4 and 5, exhibit worse performance. This performance drop is evidence that the original network contained *fragile co-adapted features* on successive layers, that is, features that interact with each other in a complex or fragile way such that this co-adaptation *could not be relearned* by the upper layers alone. Gradient descent was able to find a good solution the first time, but this was only possible because the layers were jointly trained. By layer 6 performance is nearly back to the base level, as is layer 7. As we get closer and closer to the final, 500-way softmax output layer 8, there is less to relearn, and apparently relearning these one or two layers is simple enough for gradient descent to find a good solution. Alternately, we may say that there is less co-adaptation of features between layers 6 & 7 and between 7 & 8 than between previous layers. To our knowledge it has not been previously observed in the literature that such optimization difficulties may be worse in the middle of a network than near the bottom or top.

3. The light blue BnB$^+$ points show that when the copied, lower-layer features also learn on the target dataset (which here is the same as the base dataset), performance is similar to the base case. Such fine-tuning thus prevents the performance drop observed in the BnB networks.

4. The dark red AnB diamonds show the effect we set out to measure in the first place: the transferability of features from one network to another at each layer. Layers one and two transfer almost perfectly from A to B, giving evidence that, at least for these two tasks, not only are the first-layer Gabor and color blob features general, but the second layer features are general as well. Layer three shows a slight drop, and layers 4-7 show a more significant drop in performance. Thanks to the BnB points, we can tell that this drop is from a combination of two separate effects: the drop from lost co-adaptation *and* the drop from features that are less and less general. On layers 3, 4, and 5, the first effect dominates, whereas on layers 6 and 7 the first effect diminishes and the specificity of representation dominates the drop in performance.

   Although examples of successful feature transfer have been reported elsewhere in the literature (Girshick *et al.*, 2013; Donahue *et al.*, 2013b), to our knowledge these results have been limited to noticing that transfer from a given layer is much better than the alternative of training strictly on the target task, i.e. noticing that the AnB points at some layer are much better than training all layers from scratch. We believe this is the first time that (1) the extent to which transfer is successful has been carefully quantified layer by layer, and (2) that these two separate effects have been decoupled, showing that each effect dominates in part of the regime.

5. The light red AnB$^+$ diamonds show a particularly surprising effect: that transferring features and then fine-tuning them results in networks that generalize better than those trained directly on the target dataset. Previously, the reason one might want to transfer learned features is to enable training without overfitting on small target datasets, but this new result suggests that transferring features will boost generalization performance even if the target dataset is large. Note that this effect should not be attributed to the longer total training time (450k base iterations + 450k fine-tuned iterations for AnB$^+$ vs. 450k for baseB), because the BnB$^+$ networks are also trained for the same longer length of time and do not exhibit this same performance improvement. Thus, a plausible explanation is that even after 450k iterations of fine-tuning (beginning with completely random top layers), the effects of having seen the base dataset still linger, boosting generalization performance. It is surprising that this effect lingers through so much retraining. This generalization improvement seems not to depend much on how much of the first network we keep to initialize the second network: keeping anywhere from one to seven layers produces improved performance, with slightly better performance as we keep more layers. The average boost across layers 1 to 7 is 1.6% over the base case, and the average if we keep at least five layers is 2.1%.[4] The degree of performance boost is shown in Table 1.

Table 1: Performance boost of AnB$^+$ over controls, averaged over different ranges of layers.

| layers aggregated | mean boost over baseB | mean boost over selffer BnB$^+$ |
|---|---|---|
| 1-7 | 1.6% | 1.4% |
| 3-7 | 1.8% | 1.4% |
| 5-7 | 2.1% | 1.7% |

## 4.2 Dissimilar Datasets: Splitting Man-made and Natural Classes Into Separate Datasets

As mentioned previously, the effectiveness of feature transfer is expected to decline as the base and target tasks become less similar. We test this hypothesis by comparing transfer performance on similar datasets (the random A/B splits discussed above) to that on dissimilar datasets, created by assigning man-made object classes to A and natural object classes to B. This man-made/natural split creates datasets as dissimilar as possible within the ImageNet dataset.

The upper-left subplot of Figure 3 shows the accuracy of a baseA and baseB network (white circles) and BnA and AnB networks (orange hexagons). Lines join common target tasks. The upper of the two lines contains those networks trained toward the target task containing natural categories (baseB and AnB). These networks perform better than those trained toward the man-made categories, which may be due to having only 449 classes instead of 551, or simply being an easier task, or both.

## 4.3 Random Weights

We also compare to random, untrained weights because Jarrett *et al.* (2009) showed — quite strikingly — that the combination of random convolutional filters, rectification, pooling, and local normalization can work almost as well as learned features. They reported this result on relatively small networks of two or three learned layers and on the smaller Caltech-101 dataset (Fei-Fei *et al.*, 2004). It is natural to ask whether or not the nearly optimal performance of random filters they report carries over to a deeper network trained on a larger dataset.

The upper-right subplot of Figure 3 shows the accuracy obtained when using random filters for the first $n$ layers for various choices of $n$. Performance falls off quickly in layers 1 and 2, and then drops to near-chance levels for layers 3+, which suggests that getting random weights to work in convolutional neural networks may not be as straightforward as it was for the smaller network size and smaller dataset used by Jarrett *et al.* (2009). However, the comparison is not straightforward. Whereas our networks have max pooling and local normalization on layers 1 and 2, just as Jarrett *et al.* (2009) did, we use a different nonlinearity ($\mathrm{relu}(x)$ instead of $\mathrm{abs}(\tanh(x))$), different layer sizes and number of layers, as well as other differences. Additionally, their experiment only considered two layers of random weights. The hyperparameter and architectural choices of our network collectively provide one new datapoint, but it may well be possible to tweak layer sizes and random initialization details to enable much better performance for random weights.[5]

The bottom subplot of Figure 3 shows the results of the experiments of the previous two sections after subtracting the performance of their individual base cases. These normalized performances are plotted across the number of layers $n$ that are either random or were trained on a different, base dataset. This comparison makes two things apparent. First, the transferability gap when using frozen features grows more quickly as $n$ increases for dissimilar tasks (hexagons) than similar tasks (diamonds), with a drop by the final layer for similar tasks of only 8% vs. 25% for dissimilar tasks. Second, transferring even from a distant task is better than using random filters. One possible reason this latter result may differ from Jarrett *et al.* (2009) is because their fully-trained (non-random) networks were overfitting more on the smaller Caltech-101 dataset than ours on the larger ImageNet

---

informative, however, because the performance at each layer is based on different random draws of the upper layer initialization weights. Thus, the fact that layers 5, 6, and 7 result in almost identical performance across random draws suggests that multiple runs at a given layer would result in similar performance.

[5]For example, the training loss of the network with three random layers failed to converge, producing only chance-level validation performance. Much better convergence may be possible with different hyperparameters.

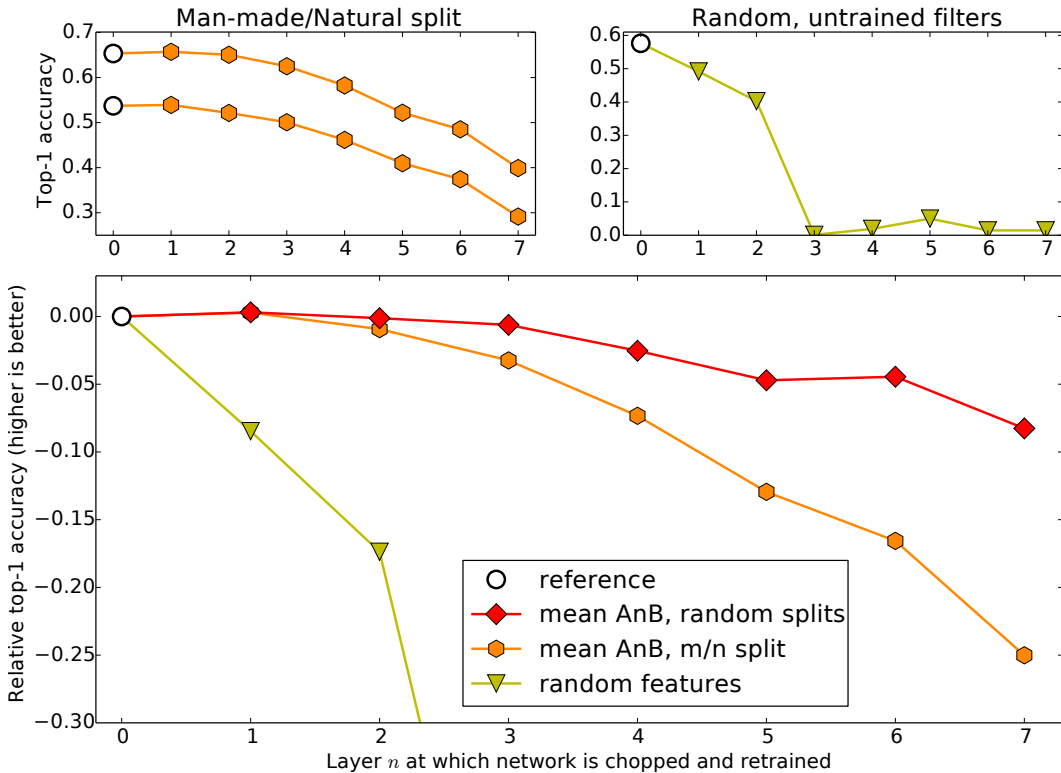

Figure 3: Performance degradation vs. layer. *Top left*: Degradation when transferring between dissimilar tasks (from man-made classes of ImageNet to natural classes or vice versa). The upper line connects networks trained to the "natural" target task, and the lower line connects those trained toward the "man-made" target task. *Top right*: Performance when the first $n$ layers consist of random, untrained weights. *Bottom*: The top two plots compared to the random A/B split from Section 4.1 (red diamonds), all normalized by subtracting their base level performance.

dataset, making their random filters perform better by comparison. In the supplementary material, we provide an extra experiment indicating the extent to which our networks are overfit.

## 5 Conclusions

We have demonstrated a method for quantifying the transferability of features from each layer of a neural network, which reveals their generality or specificity. We showed how transferability is negatively affected by two distinct issues: optimization difficulties related to splitting networks in the middle of fragilely co-adapted layers and the specialization of higher layer features to the original task at the expense of performance on the target task. We observed that either of these two issues may dominate, depending on whether features are transferred from the bottom, middle, or top of the network. We also quantified how the transferability gap grows as the distance between tasks increases, particularly when transferring higher layers, but found that even features transferred from distant tasks are better than random weights. Finally, we found that initializing with transferred features can improve generalization performance even after substantial fine-tuning on a new task, which could be a generally useful technique for improving deep neural network performance.

## Acknowledgments

The authors would like to thank Kyunghyun Cho and Thomas Fuchs for helpful discussions, Joost Huizinga, Anh Nguyen, and Roby Velez for editing, as well as funding from the NASA Space Technology Research Fellowship (JY), DARPA project W911NF-12-1-0449, NSERC, Ubisoft, and CIFAR (YB is a CIFAR Fellow).

## Footnotes

[1]The ImageNet dataset, as released in the Large Scale Visual Recognition Challenge 2012 (ILSVRC2012) (Deng *et al.*, 2009) contains 1,281,167 labeled training images and 50,000 test images, with each image labeled with one of 1000 classes.

[2]Note that $n = 8$ doesn't make sense in either case: B8B is just baseB, and A8B would not work because it is never trained on B.

[3]AnA networks and BnB networks are statistically equivalent, because in both cases a network is trained on 500 random classes. To simplify notation we label these BnB networks. Similarly, we have aggregated the statistically identical BnA and AnB networks and just call them AnB.

[4]We aggregate performance over several layers because each point is computationally expensive to obtain (9.5 days on a GPU), so at the time of publication we have few data points per layer. The aggregation is

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
