[Supplementary Material]

# Supplementary material for: How transferable are features in deep neural networks?

**Jason Yosinski,**[1]   **Jeff Clune,**[2]   **Yoshua Bengio,**[3]  and  **Hod Lipson**[4]
[1] Dept. Computer Science,   Cornell University
[2] Dept. Computer Science,   University of Wyoming
[3] Dept. Computer Science & Operations Research,   University of Montreal
[4] Dept. Mechanical & Aerospace Engineering,   Cornell University

## A   Training Details

Since Krizhevsky *et al.* (2012) won the ImageNet 2012 competition, there has naturally been much interest and work toward tweaking hyperparameters of large convolutional models. For example, Zeiler and Fergus (2013) found that it is better to decrease the first layer filters sizes from $11 \times 11$ to $7 \times 7$ and to use a smaller stride of $2$ instead of $4$. However, because this study aims not for maximum absolute performance but to use a commonly studied architecture, we used the reference implementation provided by Caffe (Jia *et al.*, 2014). We followed Donahue *et al.* (2013) in making a few minor departures from Krizhevsky *et al.* (2012) when training the convnets in this study. We skipped the data augmentation trick of adding random multiples of principle components of pixel RGB values, which produced only a $1\%$ improvement in the original paper, and instead of scaling to keep the aspect ratio and then cropping, we warped images to $256 \times 256$. We also placed the Local Response Normalization layers just *after* the pooling layers, instead of before them. As in previous studies, including Krizhevsky *et al.* (2012), we use dropout (Hinton *et al.*, 2012) on fully connected layers except for the softmax output layer.

We trained with stochastic gradient descent (SGD) with momentum. Each iteration of SGD used a batch size of 256, a momentum of 0.9, and a multiplicative weight decay (for those weights with weight decay enabled, i.e. not for frozen weights) of 0.0005 per iteration. The master learning rate started at 0.01, and annealed over the course of training by dropping by a factor of 10 every 100,000 iterations. Learning stopped after 450,000 iterations. Each iteration took about ~1.7 seconds on a NVidia K20 GPU, meaning the whole training procedure for a single network took ~9.5 days.

Our base model attains a final top-1 error on the validation set of 42.5%, about the same as the 42.9% reported by Donahue *et al.* (2013) and 1.8% worse than Krizhevsky *et al.* (2012), the latter difference probably due to the few minor training differences explained above. We checked these values only to demonstrate that the network was converging reasonably. As our goal is not to improve the state of the art, but to investigate the properties of transfer, small differences in raw performance are not of concern.

Because code is often more clear than text, we've also made all code and parameter files necessary to reproduce these experiments available on `http://yosinski.com/transfer`.

## B   How Much Does an AlexNet Architecture Overfit?

We observed relatively poor performance of random filters in an AlexNet architecture (Krizhevsky *et al.*, 2012) trained on ImageNet, which is in contrast to previously reported successes with random filters in a smaller convolutional networks trained on the smaller Caltech-101 dataset (Jarrett *et al.*, 2009). One hypothesis presented in the main paper is that this difference is observed because ImageNet is large enough to support training an AlexNet architecture without excessive overfitting. We sought to support or disprove this hypothesis by creating reduced size datasets containing the

Figure S1: Top-1 validation accuracy for networks trained on datasets containing reduced numbers of examples. The largest dataset contains the entire ILSVRC2012 (Deng *et al.*, 2009) release with a maximum of 1300 examples per class, and the smallest dataset contains only 1 example per class (1000 data points in total). *Top*: linear axes. The slope of the rightmost line segment between 1000 and 1300 is nearly zero, indicating that the amount of overfit is slight. In this region the validation accuracy rises by 0.010820 from 0.54094 to 0.55176. *Bottom*: logarithmic axes. It is interesting to note that even the networks trained on a single example per class or two examples per class manage to attain 3.8% or 4.4% accuracy, respectively. Networks trained on {5,10,25,50,100} examples per class exhibit poor convergence and attain only chance level performance.

same 1000 classes as ImageNet, but where each class contained a maximum of $n$ examples, for each $n \in \{1300, 1000, 750, 500, 250, 100, 50, 25, 10, 5, 2, 1\}$. The case of $n = 1300$ is the complete ImageNet dataset.

Because occupying a whole GPU for this long was infeasible given our available computing resources, we also devised a set of hyperparameters to allow faster learning by boosting the learning rate by 25% to 0.0125, annealing by a factor of 10 after only 64,000 iterations, and stopping after 200,000 iterations. These selections were made after looking at the learning curves for the base case and estimating at which points learning had plateaued and thus annealing could take place. This faster training schedule was only used for the experiments in this section. Each run took just over 4 days on a K20 GPU.

The results of this experiment are shown in Figure S1 and Table S1. The rightmost few points in the top subplot of Figure S1 appear to converge, or nearly converge, to an asymptote, suggesting that validation accuracy would not improve significantly when using an AlexNet model with much more data, and thus, that the degree of overfit is not severe.

Table S1: An enumeration of the points in Figure S1 for clarity.

| Number of examples per class | Top-1 validation accuracy |
|---:|---|
| 1300 | 0.55176 |
| 1000 | 0.54094 |
| 750 | 0.51470 |
| 500 | 0.47568 |
| 250 | 0.38428 |
| 100 | 0.00110 |
| 50 | 0.00111 |
| 25 | 0.00107 |
| 10 | 0.00106 |
| 5 | 0.00108 |
| 2 | 0.00444 |
| 1 | 0.00379 |

## C  Man-made vs. Natural Split

In order to compare transfer performance between tasks A and B such that A and B are as semantically dissimilar as possible, we sought to find two disjoint subsets of the 1000 classes in ImageNet that were as unrelated as possible. To this end we annotated each node $x_i$ in the WordNet graph with a label $n_i$ such that $n_i$ is the number of distinct ImageNet classes reachable by starting at $x_i$ and traversing the graph only in the parent $\rightarrow$ child direction. The 20 nodes with largest $n_i$ are the following:

```
 n_i    x_i
1000    n00001740: entity
 997    n00001930: physical entity
 958    n00002684: object, physical object
 949    n00003553: whole, unit
 522    n00021939: artifact, artefact
 410    n00004475: organism, being
 410    n00004258: living thing, animate thing
 398    n00015388: animal, animate being, beast, brute, creature, fauna
 358    n03575240: instrumentality, instrumentation
 337    n01471682: vertebrate, craniate
 337    n01466257: chordate
 218    n01861778: mammal, mammalian
 212    n01886756: placental, placental mammal, eutherian, eutherian mammal
 158    n02075296: carnivore
 130    n03183080: device
 130    n02083346: canine, canid
 123    n01317541: domestic animal, domesticated animal
 118    n02084071: dog, domestic dog, Canis familiaris
 100    n03094503: container
  90    n03122748: covering
```

Starting from the top, we can see that the largest subset, `entity`, contains all 1000 ImageNet categories. Moving down several items, the first subset we encounter containing approximately half of the classes is `artifact` with 522 classes. The next is `organism` with 410. Fortunately for this study, it just so happens that these two subsets are mutually exclusive, so we used the first to populate our *man-made* category and the second to populate our *natural* category. There are $1000 - 522 - 410 = 68$ classes remaining outside these two subsets, and we manually assigned these to either category as seemed more appropriate. For example, we placed `pizza`, `cup`, and `bagel` into *man-made* and `strawberry`, `volcano`, and `banana` into *natural*. This process results in 551 and 449 classes, respectively. The 68 manual decisions are shown below, and the complete list of 551 man-made and 449 natural classes is available at `http://yosinski.com/transfer`.

Classes manually placed into the man-made category:

```
n07697537 hotdog, hot dog, red hot
n07860988 dough
n07875152 potpie
n07583066 guacamole
n07892512 red wine
n07614500 ice cream, icecream
n09229709 bubble
n07831146 carbonara
n07565083 menu
n07871810 meat loaf, meatloaf
n07693725 bagel, beigel
n07920052 espresso
n07590611 hot pot, hotpot
n07873807 pizza, pizza pie
n07579787 plate
n06874185 traffic light, traffic signal, stoplight
n07836838 chocolate sauce, chocolate syrup
n15075141 toilet tissue, toilet paper, bathroom tissue
n07613480 trifle
n07880968 burrito
n06794110 street sign
n07711569 mashed potato
n07932039 eggnog
n07695742 pretzel
n07684084 French loaf
n07697313 cheeseburger
n07615774 ice lolly, lolly, lollipop, popsicle
n07584110 consomme
n07930864 cup
```

Classes manually placed into the natural category:

```
n13133613 ear, spike, capitulum
n07745940 strawberry
n07714571 head cabbage
n09428293 seashore, coast, seacoast, sea-coast
n07753113 fig
n07753275 pineapple, ananas
n07730033 cardoon
n07749582 lemon
n07742313 Granny Smith
n12768682 buckeye, horse chestnut, conker
n07734744 mushroom
n09246464 cliff, drop, drop-off
n11879895 rapeseed
n07718472 cucumber, cuke
n09468604 valley, vale
n07802026 hay
n09288635 geyser
n07720875 bell pepper
n07760859 custard apple
n07716358 zucchini, courgette
n09332890 lakeside, lakeshore
n09193705 alp
n09399592 promontory, headland, head, foreland
n07717410 acorn squash
n07717556 butternut squash
n07714990 broccoli
n09256479 coral reef
n09472597 volcano
n07747607 orange
n07716906 spaghetti squash
n12620546 hip, rose hip, rosehip
n07768694 pomegranate
n12267677 acorn
n12144580 corn
n07718747 artichoke, globe artichoke
n07753592 banana
n09421951 sandbar, sand bar
n07715103 cauliflower
n07754684 jackfruit, jak, jack
```

## Supplementary References

Deng, J., Dong, W., Socher, R., Li, L.-J., Li, K., and Fei-Fei, L. (2009). ImageNet: A Large-Scale Hierarchical Image Database. In *CVPR09*.

Donahue, J., Jia, Y., Vinyals, O., Hoffman, J., Zhang, N., Tzeng, E., and Darrell, T. (2013). Decaf: A deep convolutional activation feature for generic visual recognition. Technical report, arXiv preprint arXiv:1310.1531.

Hinton, G. E., Srivastava, N., Krizhevsky, A., Sutskever, I., and Salakhutdinov, R. (2012). Improving neural networks by preventing co-adaptation of feature detectors. Technical report, arXiv:1207.0580.

Jarrett, K., Kavukcuoglu, K., Ranzato, M., and LeCun, Y. (2009). What is the best multi-stage architecture for object recognition? In *Proc. International Conference on Computer Vision (ICCV'09)*, pages 2146–2153. IEEE.

Jia, Y., Shelhamer, E., Donahue, J., Karayev, S., Long, J., Girshick, R., Guadarrama, S., and Darrell, T. (2014). Caffe: Convolutional architecture for fast feature embedding. *arXiv preprint arXiv:1408.5093*.

Krizhevsky, A., Sutskever, I., and Hinton, G. (2012). ImageNet classification with deep convolutional neural networks. In *Advances in Neural Information Processing Systems 25 (NIPS'2012)*.

Zeiler, M. D. and Fergus, R. (2013). Visualizing and understanding convolutional networks. Technical Report Arxiv 1311.2901.