[Reviews · NeurIPS 2014]

Submitted by Assigned_Reviewer_13

This paper aims to quantify the transferrability of features in deep neural networks, both in terms of the difference between source and target tasks and in terms of the depth of the features being transferred. To this end, the authors take an existing network (Krizhevsky et al. 2012), and performs generalization by fixing different layer depth and by transferring between different splits of the ImageNet dataset.

I find the paper sufficiently interesting in the sense that, despite the many papers describing the success of feature transfer (e.g. Decaf, deconvnet, overfeat), we still lack a principled way to analyze why, and to what level, such feature transfer may be successful. The paper provides a reasonable effort in doing so. Another interesting finding is that transfer + fine-tuning almost always helps, even if the target task has sufficient data to train classifiers from scratch (see Figure 2).

On the other side, this paper seems a bit empirical, in the sense that experiments are carried out on some manual decisions (such as split of source and target tasks). It would be good to see a quantitative criterion analyzing the correlation between the two factors raised in the paper and the final performance. Of course, the authors have fairly acknowledged so, and have not overclaimed anything beyond what is presented.

The reviewer does not have a strong opinion on the paper, but believe that the analysis presented in the paper may benefit vision practitioners, especially those who would like to apply deep neural networks in a transfer learning fashion.
Summary: This paper provides an experimental analysis on the transferrability of features in deep neural networks, using the de-facto standard benchmark of ImageNet and Krizhevsky's 2012 model.

Submitted by Assigned_Reviewer_35

This paper systematically studies the transferrability of deep convnet features across a few experimental settings.
The experiments are focused on the "AlexNet" architecture and the ILSVRC-2012 dataset, with the main set of results on transfer learning problems induced by dividing the 1000 ILSVRC-2012 object classes into random 500/500 splits. The N-layer network is trained on split A, then layers 1 through k are copied to a new network and layers k+1 through N are randomly initialized and retrained on split B.

Results are reported across both choices of allowing layers 1 through k to be fine-tuned vs. frozen, and across all choices of k.
Another set of results shows these in the setting of a manually chosen split of natural vs. man-made classes, and yet another set of results explores using random, un-trained weights in some layers.
Many interesting observations are made which were not previously reported, particularly results #3 and #5 in section 4.1.

Despite the large number of papers that either extract features from or fine-tune the ImageNet-pretrained "AlexNet" CNN, this is the first work I know of to do a clean, thorough exploration of the space of choices one has to make when using this class of methods.

While the paper offers little in the way of technical or mathematical novelty, it does not strive to -- it provides an extensive, clean set of numerical answers to many commonly asked questions, and I believe its results give valuable practical insight to researchers and practitioners of modern deep learning methods.

The authors also promise to release the code they use to run these experiments, allowing other researchers to explore how these experiments generalize to other architectures, etc.
Summary: This paper systematically explores the transferrability of deep CNN features depending on the fine-tuning protocol and the layer of the network used. It does not propose any new method, but offers an extensive set of reference results and plenty of valuable insight about the "how" and "why" of transfer learning using deep networks

Submitted by Assigned_Reviewer_40

This paper addresses the transferability of features in deep networks when trained on some data and tested on other, in a systematic way. In a first stage, two deep networks are trained on two separate halves of ImageNet. Then the networks are split at all possible intermediate networks, and various cross-training schemes are compared (e.g., freezing the bottom part, or training the whole network, on the same or a different part of ImageNet). 
A set of interesting conclusions are presented; e.g., joint training of a network  leads to co-adaptation if too many layers are kept frozen, so that performance decreases even if the same part of the training set is used to re-train. Cross-training between two different training sets and keeping too much of the original network leads to even more loss of performance when too many layers are kept. And an exciting result is that pre-training on another training set, then training the whole network on the final training set, gives the best performance of all.

The paper is well written and very clear. The thoroughness of the approach is appealing. While this paper does not improve the state of the art, it gives valuable insight into what happens when a network is switched from one training set to another. It is also great to know that the code for this paper will be released.
Summary: This well-written paper thoroughly explores how pre-training on a given training set and transferring to another training set is affected by the depth of the pre-trained network, the training procedure, and the similarity of training sets. 
The contribution is focused but valuable, and I believe this systematic set of experiments would be of interest to many researchers.
Author Feedback
Author rebuttal: We would like to thank the reviewers for their encouraging feedback and insightful comments. Almost all of the reviewer comments were positive. Below we reply to one comment and the one suggestion for a potential change to the paper.

REVIEWER 1

> On the other side, this paper seems a bit empirical, in
> the sense that experiments are carried out on some manual
> decisions (such as split of source and target tasks).

Admittedly, it was our choice to perform separation into natural vs. man-made subsets. However, given the hierarchical class structure of WordNet and ImageNet, these two subsets were the only two large exclusive subsets available. This can be seen in the below list of the top 14 subsets of classes, sorted by number of classes (first column) in each subset:

158 n02075296 - carnivore
212 n01886756 - placental, placental mammal, eutherian, eutherian mammal
218 n01861778 - mammal, mammalian
337 n01466257 - chordate
337 n01471682 - vertebrate, craniate
358 n03575240 - instrumentality, instrumentation
398 n00015388 - animal, animate being, beast, brute, creature, fauna
410 n00004258 - living thing, animate thing
410 n00004475 - organism, being
522 n00021939 - artifact, artefact
949 n00003553 - whole, unit
958 n00002684 - object, physical object
997 n00001930 - physical entity
1000 n00001740 - entity

Starting at the largest subset "1000 n00001740 - entity" containing all 1000 classes and working backward, the first subset containing approximately 50% of the classes is "522 n00021939 - artifact, artefact", and the next is "410 n00004475 - organism, being". Fortunately for this study, these two subsets are mutually exclusive; the first became our "man-made" category and the second our "natural" category. Our only real influence was in choosing what to do with the remaining 1000-522-410=68 classes (Are cooked squash natural or man-made?). We looked at example images from each of the 68 classes and placed them into either "natural" or "man-made" as we thought best. In our code release, we provide a list of the 551 vs 449 classes in each category, and we highlight those 68 category decisions that were made manually.

We initially left most of this discussion out of the paper to save space, but thanks to this comment, we realize that in its absence the decisions do seem arbitrary. Thus, we will include this discussion in the final version.

> It would be good to see a quantitative criterion analyzing
> the correlation between the two factors raised in the
> paper and the final performance. Of course, the authors
> have fairly acknowledged so, and have not overclaimed
> anything beyond what is presented.

This is an interesting idea, though we're not sure exactly what sort of quantitative criterion would be most helpful. Could you elaborate on what you envision? We have spent some time discussing how best to implement your suggested analysis, but we concluded that we do not yet understand your suggestion enough to implement it precisely. We would be happy to include such an analysis once we understand exactly what is being suggested.

GENERAL

One additional change we plan to make, provided the reviewers do not object, is to add even more replicates of as many experiments as possible to increase the statistical strength of our results. Trials are ongoing, and we will add as many data points as our available computation allows.

SUMMARY

To summarize, we plan to make the following changes for a final version of the paper:

- Add discussion of motivation behind creation of natural vs. man-made split, suggested by Reviewer 1.

- Possibly add model suggested by Reviewer 1.

- Strengthen statistical conclusions by averaging over more randomly initialized trials.